# Selective Proliferation of Highly Functional Adipose-Derived Stem Cells in Microgravity Culture with Stirred Microspheres

**DOI:** 10.3390/cells10030560

**Published:** 2021-03-04

**Authors:** Takanobu Mashiko, Koji Kanayama, Natsumi Saito, Takako Shirado, Rintaro Asahi, Masanori Mori, Kotaro Yoshimura

**Affiliations:** 1Department of Plastic Surgery, Jichi Medical University 3311-1, Yakushiji, Shimotsuke-Shi, Tochigi 329-0498, Japan; takanobu-mashiko@umin.ac.jp (T.M.); kanayama-tky@umin.ac.jp (K.K.); natsaito@jichi.ac.jp (N.S.); shirado@jichi.ac.jp (T.S.); asasihitohito@yahoo.co.jp (R.A.); mamori-kyt@umin.ac.jp (M.M.); 2Department of Plastic Surgery, Toranomon Hospital 2-2-2, Toranomon, Minato-Ku, Tokyo 105-8470, Japan

**Keywords:** adipose-derived stem cell, microgravity culture, polystyrene microsphere, collagen microsphere, multilineage-differentiating stress-enduring cell

## Abstract

Therapeutic effects of adult stem-cell transplantations are limited by poor cell-retention in target organs, and a reduced potential for optimal cell differentiation compared to embryonic stem cells. However, contemporary studies have indicated heterogeneity within adult stem-cell pools, and a novel culturing technique may address these limitations by selecting those for cell proliferation which are highly functional. Here, we report the preservation of stemness in human adipose-derived stem cells (hASCs) by using microgravity conditions combined with microspheres in a stirred suspension. The cells were bound to microspheres (100−300 μm) and cultured using a wave-stirring shaker. One-week cultures using polystyrene and collagen microspheres increased the proportions of SSEA-3(+) hASCs 4.4- and 4.3-fold (2.7- and 2.9-fold increases in their numbers), respectively, compared to normal culture conditions. These cultured hASCs expressed higher levels of pluripotent markers (OCT4, SOX2, NANOG, MYC, and KLF), and had improved abilities for proliferation, colony formation, network formation, and multiple-mesenchymal differentiation. We believe that this novel culturing method may further enhance regenerative therapies using hASCs.

## 1. Introduction

Human adipose-derived stem cells (hASCs) are easily isolated therapeutic stem cells from adipose tissue. They have increasingly been used to ameliorate a broad range of refractory diseases, including irradiated tissue [1], ischemic wounds [2], and diabetic ulcers [3]. However, their clinical efficacy has been limited because the long-term survival of transplanted hASCs is diminished by environmental stress [4,5,6]. Consequently, hASC therapeutic mechanisms have been limited to the secretion of cytokines and growth factors to promote cell recruitment, immunomodulation, extracellular matrix remodeling, and angiogenesis [7,8,9,10]. In contrast, embryonic stem cells retain their potency and are thought to be superior to adult stem cells for regenerative potential [11]. However, the use of embryonic stem cells is limited by ethical issues [12].

Modern research has made gains toward a better understanding of stem-cell heterogeneity [13,14]. Within stem-cell pools (previously thought to be homogenous), a surprising degree of subpopulation dynamics in gene expression has been revealed related to tissue regeneration, cell survival, proliferation, and stemness [15]. This concept of functional individuality within a stem-cell pool is especially important for hASCs, because cells with higher functional properties would be expected to have more promising therapeutic benefits through an increased capacity to survive under difficult conditions, to differentiate into various mesenchymal cell types, and to home into damaged tissues [16,17,18]. One example of a highly functional hASC candidate is the multilineage-differentiating stress-enduring (Muse) cell, which can be isolated from hASCs using its expression of stage-specific embryonic antigen-3 (SSEA-3) [19] and has been reported to have novel potential for liver cirrhosis [20], brain infarction [21], and aortic aneurysm [22] treatments.

An additional problem is that the expansion of such highly-pluripotent cell populations is quite difficult. Adhesion-based cultures, the standard technique for in vitro expansion of hASCs, are known to lead to a gradual quiescence in stemness indicated, for example, by a decline in CD34 positivity [23]. In contrast, suspension-type cultures have also been used successfully for temporary cultivations of Muse cells, and although the mechanism is unknown, it has been reported that a microgravity (i.e., weightless) environment had a positive effect on stem-cell cultures for maintenance of their original potencies [24]. However, the forced-floating of hASC cultures using non-adherent dishes caused spherical aggregations and oxygen/nutrient diffusion limits, which resulted in central necrosis in hASC spheroids [25]. There is, therefore, an urgent need for new and advanced culture techniques to permit the selective expansion of hASC subpopulations with improved properties for proliferation, differentiation, angiogenesis, immunosuppression, and stress-resistance.

The microcarrier system was first introduced in 1967 for the growth of fibroblasts [26], and has been used for the expansion of other adherent cells such as chondrocytes [27], hepatocytes [28], neural stem cells [29], and hASCs [30,31]. Microcarriers are usually small spherical particles which can adhere to cultivated cells and are characterized by higher surface-to-volume ratios that can accommodate higher cell densities compared to monolayer cultures. Most importantly, under the dynamic conditions of continuous shaking, microcarrier culturing has a multidirectional gravity effect on the adhered cells, effectively mimicking floating cultures. Therefore, the aim of this study was to evaluate the ability of dynamic microgravity culturing of hASCs using microcarriers to selectively expand highly functional stem-cell subpopulations.

## 2. Materials and Methods

### 2.1. Primary Culture of hASCs

Human lipoaspirates were obtained from 12 healthy female donors, average age 42.2 ± 9.3 years (mean ± SD) and average body mass index of 22.4 ± 1.8 kg/m^2^, who underwent abdominal liposuction. Each patient provided written informed consent, and the research protocol was approved by the institutional review board. The stromal vascular fraction (SVF) was isolated from the lipoaspirates as described previously [23]. Briefly, aspirated fat tissue was digested using 0.075% collagenase (Wako Pure Chemical Industries, Osaka, Japan) at 37 °C for 30 min on a shaker. After centrifugation (800× *g* for 10 min), SVF cells were obtained as cellular pellets, filtered through 100, 70, and 40 μm mesh, and resuspended. The SVF cells were plated at a density of 5 × 10^5^ nucleated cells/100 mm dish and cultured at 37 °C in an atmosphere of 5% carbon dioxide in humid air. Cells were grown in Dulbecco’s Modified Eagle Medium with Ham’s F-12 (DMEM/F12; Wako Pure Chemical Industries) supplemented with 10% fetal bovine serum. Primary cells were cultured until near confluence. Then, the adherent cells were released using a proteolytic enzyme treatment (TrypLE Express, Invitrogen, Carlsbad, CA, USA), defined as passage 0 (P0) hASCs, and transferred to another dish. Once the adherent hASCs reached 80% confluency, cells were passaged using TrypLE Express, and P3 hASCs were used in the following experiments. Using a multicolor flow cytometer (MACS-Quant, Miltenyi Biotec, Bergisch Gladbach, Germany), P3 hASCs were characterized for the positive expressions of CD73, CD90, and CD105 and the negative expression of CD45 before use.

### 2.2. Characterization of SSEA-3-Positive Cells in the hASC Pool

First, SSEA-3 marker expression was assessed in normal hASCs using flow cytometry. Adherent P3 hASCs from each donor were detached using TrypLE Express, centrifuged, and washed with phosphate-buffered saline (PBS). The cells were sieved through 100 and 40 μm mesh filters, pelleted by centrifugation, and then resuspended for analysis. The isolated cells were incubated with rat anti-SSEA-3 antibody (1:50; BioLegend, San Diego, CA, USA) and detected using an fluorescein isothiocyanate-conjugated anti-rat IgM (BD Biosciences, San Diego, CA, USA). Analyses were performed using a multicolor flow cytometer (MACS-Quant). Control gates were set based on staining with a labeled non-specific antibody (matched isotype control immunoglobulin G (IgG)); no more than 0.1% of the cells were deemed positive using the non-specific antibody.

To assess the stress durability of SSEA-3-positive cells, adherent P3 hASCs were exposed to a variety of optimized stress conditions (heat, 45 °C for 1 min; low-pH solution (pH = 5) for 60 min; proteolysis, TrypLE Express for 20 h at 37 °C; hypotonia, Milli-Q water for 1 min; and mechanostress, transferred 30 times between two syringes through a connector with a small hole). One day after stress exposure, cell numbers and viability were measured using a dual-fluorescence automated cell counter (Luna-FL, Logos Biosystems, Gyeonggi-do, Korea), and SSEA-3 positivity was detected by flow cytometry after gating dead cells from live cells using 7-amino-actinomycin D (7AAD, BD Biosciences) fluorescence (*n* = 12).

### 2.3. Preparation of Microspheres

Two kinds of microspheres were used as cell carriers during cell culture. First, crosslinked polystyrene microspheres (Polystyrene Beads Large, Polysciences, Warrington, PA, USA) with diameters between 200−300 μm were used. For good cell attachment, the polystyrene surfaces were hydrophilized using a 30 min plasma treatment from a vacuum plasma apparatus (YHS-DφS, Sakigake-Semiconductor, Kyoto, Japan). Ten grams of polystyrene microspheres (approximately 2.4 million microbeads with a total surface area of 4000 cm^2^) were washed three times in 70% ethanol for sterilization and diluted in 15 mL of PBS. Secondly, collagen microspheres (100−200 μm diameters, Cellagen; Koken, Tokyo, Japan) manufactured from reconstituted collagen obtained from bovine skin and crosslinked with 0.5% hexamethylene diisocyanate were used. Pre-sterilized vials (15 mL) contained approximately three million collagen beads with a total surface area of approximately 4000 cm^2^.

### 2.4. hASC Loading onto Microspheres for Microgravity Culture

The two types of microspheres were washed with PBS and resuspended in warm culture medium overnight (DMEM/F12) prior to use. Polystyrene and collagen microsphere solutions (0.3 mL), representing a total surface area of approximately 80 cm^2^, were added to 3.5 cm diameter, non-adherent, polystyrene dishes (Corning Inc., Corning, NY, USA). The dishes were placed onto a continuous-action shaker (Wave-PR, Taitec, Saitama, Japan), capable of shaking the microspheres in three dimensions by combining movements from two axes (rotary and seesaw). As a result, the cumulative gravity vector toward the microspheres was minimized towards zero (i.e., microgravity). The unit was driven by a computer system which allowed for the adjustment of rotation speed and shaking angle (see Appendix A).

For each dish on the shaker, 1 × 10^5^ P3 hASCs were suspended in 300 μL of DMEM/F12 and incubated at 37 °C in an atmosphere of 5% carbon dioxide in humid air. After allowing the mixture of cells and microcarriers to settle for 30 min, 2 min of shaking was started at a setting of 10 rpm and a shaking angle of 6 degrees, and then stopped for a further 28 min. This 30 min procedure was repeated eight times (4 h total) to allow the cells to adhere to the microspheres circumferentially. Then, DMEM/F12 was added (1.5 mL) and the suspension was stirred continuously at 15 rpm at a 6° shaking angle. Using this continuous stirring, the cells experienced a microgravity environment while being strongly attached to the microspheres. Culturing continued for up to two weeks, and the culture medium was changed twice per week as follows; after allowing the microspheres to settle, half of the supernatant was exchanged for fresh medium.

### 2.5. Cell Retrieval from Microspheres

When ready for analysis, the cell–microcarrier complexes obtained from the culture dishes were transferred to new tubes (15 mL) and sedimented. For each tube, the supernatant was removed, and the complexes were washed using PBS. Cells were enzymatically released by incubation with 5 mL of TrypLE Express for 15 min at 37 °C. Cell suspensions were diluted with additional culture medium (5 mL) and sieved through 100 and 40 μm mesh collector screens. Cells were pelleted by centrifugation at 600× *g* for 5 min, and then resuspended for analysis. No microcarriers (fractured or intact) were seen in the retrieved cell suspensions.

### 2.6. Cell Growth in Microsphere Cultures

Numerical measurement of viable cells cultured on polystyrene and collagen microspheres, or in polystyrene and collagen dishes was carried out using a dual-fluorescence automated cell counter (Luna-FL). P3 hASCs (1 × 10^5^) were seeded onto polystyrene and collagen microspheres under dynamic conditions, or onto polystyrene and collagen dishes as static monolayer cultures. The cells were retrieved by trypsinization and counted on days 1, 3, 5, 7, and 14, and growth rates were calculated (i.e., hASC growth rate = hASC number on day 7/day 1). Data were gathered from six separate samples. Visualizations of hASCs cultured on polystyrene and collagen microspheres were carried out on day 7 using a light microscope (Leica DM-LB, Wetzlar, Germany) and a scanning electron microscope (SEM; S-3500N, Hitachi, Tokyo, Japan) at a beam energy of 25 kV. For the observing the collagen microspheres, a cryo-SEM technique at −130 °C was used so as not to scatter and lose uncharged microspheres.

### 2.7. Proliferation of SSEA-3-Positive hASCs in Microsphere Cultures

To determine whether microsphere culturing and/or dynamic conditions enhanced the proliferation of highly functional hASCs, we assessed cell number, viability, and SSEA-3-positivity using flow cytometry in six groups: hASCs retrieved from either polystyrene or collagen microcarriers under dynamic conditions, those retrieved from either polystyrene or collagen microcarriers under static conditions, and those retrieved from either polystyrene or collagen dishes. The number of SSEA-3(+) cells was measured after seven days of culture under each condition, and growth rates were calculated (i.e., SSEA-3(+) growth rate = SSEA-3(+) cell number on day 7/day 0). Six separate samples were used for the data analysis.

### 2.8. Immunocytochemistry

P3 hASCs were cultured for seven days on polystyrene or collagen microspheres, and in polystyrene or collagen dishes, fixed with 4% paraformaldehyde, washed in PBS, and permeabilized with 0.2% Triton X-100 (Sigma-Aldrich, St. Louis, MO, USA). Cells were incubated for 16 h (4 °C) with the following primary antibodies: anti-OCT4 (GeneTex, Irvine, CA, USA); anti-SOX2 (GeneTex); anti-NANOG (N3C3, GeneTex); and anti-SSEA-3 (EMD Millipore, Darmstadt, Germany). All primary antibodies were diluted 1:200 in a solution of PBS/0.1% bovine serum albumin. Appropriate secondary antibodies (1:200 dilutions of either goat anti-rabbit IgG Alexa Fluor 594, goat anti-rabbit IgG Alexa Fluor 488, or goat anti-rat Alexa Fluor 594) were incubated with cells for 1 h at room temperature. Finally, cells were washed with PBS and treated with Hoechst 33342 (1:500; Dojindo, Kumamoto, Japan) for 10 min. Stained cells were visualized as 30 μm Z-stack fluorescence images collected in 1.5 μm steps taken by a fluorescence-equipped microscope (Keyence, Osaka, Japan).

### 2.9. Quantitative Real-Time Polymerase Chain Reaction (RT-PCR)

Total RNA was isolated from hASCs cultured with either polystyrene or collagen microspheres under dynamic conditions, polystyrene or collagen microspheres under static conditions, and from those cultured on normal polystyrene dishes (control) for seven days using an RNeasy Mini kit (Qiagen, Carlsbad, CA, USA), followed by reverse transcription. RT-PCR was performed using a StepOnePlus RT-PCR system (Thermo Fisher Scientific, Waltham, MA, USA) using a fast SYBR Green PCR master mix (Thermo Fisher Scientific) and the primers listed in Table 1. Expression levels were calculated by a comparative CT method relative to a common endogenous reference gene, *ACTB*. Data were taken from three separate samples.

### 2.10. Colony-Forming Assay

P3 hASCs were cultured either with polystyrene or collagen microspheres under dynamic conditions or in polystyrene dishes for seven days, and then seeded (1 × 10^2^ cells) into polystyrene 6-well plates. Cells were incubated at 37 °C for ten days, fixed with methanol, stained with cresyl violet solution, and then counted using digital imaging software (Photoshop; Adobe Systems Inc., San Jose, CA, USA) (*n* = 6).

### 2.11. In Vitro Angiogenesis (Network Formation) Assay

P3 hASCs were cultured either with polystyrene or collagen microspheres under dynamic conditions or in polystyrene dishes for seven days, and then cell-network formation was assessed. Matrigel (BD Biosciences) was applied to 96-well plates (50 μL per well) and polymerized (30 min at 37 °C). After incubation in endothelial basal medium (EBM; Cambrex, Walkersville, MD, USA) containing 2% fetal bovine serum for 24 h as a pretreatment, 5000 cells (resuspended in 50 μL of EBM) were plated into the Matrigel wells and cultured for 6 h. Branch formation was observed with phase microscopy, and the lengths of cytoplasmic extensions per field were calculated (*n* = 6).

### 2.12. Multilineage Differentiation Assay

P3 hASCs were cultured either with polystyrene or collagen microspheres under dynamic conditions or in polystyrene dishes for seven days, and then differentiated into adipogenic, osteogenic, and chondrogenic lineages (*n* = 3). Cells were seeded into polystyrene 6-well plates and incubated in DMEM/F12 until confluent. Adipogenic/osteogenic differentiation was initiated using either adipogenic medium (Adipolife; Lifeline Cell Technology, Frederick, MD, USA) or osteogenic medium (Osteolife; Lifeline Cell Technology). After culturing for 21 days (medium changed every three days), adipogenic ability was analyzed qualitatively with oil red O staining and quantitatively with AdipoRed reagent (Lonza, Basel, Switzerland). Osteogenic ability was analyzed qualitatively using von Kossa staining and quantitatively by Calcium-E testing (Wako Pure Chemicals). Chondrogenic differentiation was conducted using a micromass culture system; hASCs were resuspended in chondrogenic medium (Chondrolife; Lifeline Cell Technology) and centrifuged (800× *g* for 5 min). Half of the medium was changed after centrifugation every two days. After two weeks, chondrogenic ability was analyzed qualitatively using Alcian blue staining and quantitatively by measuring micromass diameters.

### 2.13. Statistics

Data were analyzed using SPSS package 23.0 (SPSS, Inc., Chicago, IL, USA) and Kyplot 2.0 (Freeware). Based on the Kolmogorov–Smirnov test, the data were normally distributed within the donor populations. Therefore, a one-way analysis of variance (ANOVA) and Bonferroni adjustments for post-hoc comparisons were used to determine significant differences between means (*p* < 0.05 representing significance, unless otherwise stated).

## 3. Results

### 3.1. Characterization of SSEA-3-Positive hASCs

Cells were expanded for three passages before characterization using flow cytometry and combinations of mesenchymal-cell surface markers. We found that 94.9 ± 3.3% of cells were positive for CD105, CD90, and CD73, and that 99.5 ± 0.3% were negative for CD45. Then, the total number of cells, their viability, and SSEA-3-positivity of viable cells were assessed in normal hASCs (control) and in five kinds of stress-treated hASCs. One day after stress exposure, even though all stressed groups showed significant reductions (62.4–74.2%) in viable cells compared to the control group, stressed cells exhibited higher proportions of SSEA-3(+) cells (heat, 2.1%; low pH, 3.1%; proteolysis, 4.2%; hypotonia, 2.2%; mechanostress, 1.4%) compared to controls (1.3%) (Figure 1A). No significant differences were detected in the overall numbers of SSEA-3(+) cells between stress-treated hASCs and normal hASCs, with the exception of the mechanostress group, which showed a significantly lower number of SSEA-3(+) cells (*p* < 0.01) (Figure 1B,C).

### 3.2. Cell Growth in Microgravity Microsphere Cultures

Adherent cells are continuously affected by the downward force of gravity throughout the culture period, but cells attached to continuously stirred microspheres are affected by multidirectional gravitational forces (i.e., microgravity) (Figure 2A). The numbers of hASCs were decreased in both polystyrene and collagen microsphere cultures on day 1, suggesting that some cells did not successfully adhere to the microbeads (Figure 2B). However, the growth rates (i.e., hASC number on day 7/day 1) used to estimate true cell proliferation showed no significant differences between groups (polystyrene microsphere, 6.0; collagen microsphere, 6.0; polystyrene dish, 5.7; collagen dish, 5.8). On day 14, there were no significant differences in the numbers of hASCs retrieved from each culture, possibly because the monolayer cultures reached confluency. Visualizations of hASCs that were adhered to polystyrene and collagen microspheres at culture-day 7 revealed their successful proliferation (Figure 2C).

### 3.3. Proliferation of SSEA-3-Positive hASCs in Microgravity/Microsphere Cultures

As described above, the numbers of viable hASCs on day 7 were lower in the microsphere-cultured groups, regardless of the material (polystyrene or collagen) or the condition (dynamic or static), compared to the dish-cultured groups (Figure 3A). However, when compared to polystyrene monolayer cultures (as controls), the static-condition polystyrene and collagen microsphere cultures produced 2.3-fold and 2.2-fold proportional increases, and 1.2-fold and 1.3-fold increases in the numbers of SSEA-3(+) hASCs, respectively (*p* < 0.05 for both). In addition, the dynamic-culture conditions produced 4.4-fold and 4.3-fold proportional increases, and 2.7-fold and 2.9-fold increases in the numbers of SSEA-3(+) hASCs compared to the dish cultures, respectively (*p* < 0.001 for both) (Figure 3B,C). These results indicate positive effects for both the microsphere and microgravity conditions on SSEA-3(+) cell proliferation and on SSEA-3(+) cell growth rates (i.e., growth rate = SSEA-3(+) cell number on day 7/day 0) Table 2.

### 3.4. Immunocytochemistry

All of the pluripotent stem-cell markers examined were expressed in hASCs cultured in microgravity/microsphere cultures (Figure 4). The images represent those fields where the most significant differences between the dish groups and the microsphere groups were detected. Notably, hASCs cultured on polystyrene microcarriers showed strong OCT4 and SOX2 staining, while hASCs cultured on collagen microcarriers showed strong NANOG and SSEA-3 staining. By comparison, the hASCs cultured in both polystyrene and collagen monolayer cultures stained either negative, or weakly positive, for these stemness markers.

### 3.5. Selected Gene Expression Analysis by RT-PCR

We used RT-PCR to compare the expression of several genes involved with stemness and mesenchymal differentiation in hASCs from microgravity/microsphere cultures to their expressions in hASCs from dish cultures (*n* = 3). The hASCs cultured with polystyrene microspheres under dynamic conditions exhibited significantly upregulated *OCT4*, *SOX2*, *KLF4*, and *CD34* expressions, while hASCs cultured with collagen microspheres under dynamic conditions showed up-regulations of *SOX2*, *NANOG*, *MYC*, *KLF4*, and *CD34* expressions (Figure 5). In contrast, hASCs from microsphere cultures under static conditions (both polystyrene and collagen) displayed no statistically significant differences for all the genes examined.

### 3.6. Colony-Forming Assay

After 10 days of culture with an initial seeding of 1 × 10^2^ cells per well, we counted cell colonies as an estimate of proliferative ability. We found a significantly higher number of colonies in the polystyrene microsphere group (*p* = 0.027) and in the collagen microsphere group (*p* = 0.025) compared to the polystyrene dish group (*n* = 6) (Figure 6).

### 3.7. In Vitro Angiogenesis Assay

The network-formation assay showed that hASCs cultured with polystyrene and collagen microspheres formed complex, capillary-like networks more quickly and intricately than dish-cultured hASCs (Figure 7A). Accordingly, the lengths of cytoplasmic extensions were also significantly longer in the polystyrene- and collagen-microsphere groups compared to the dish group (*p* < 0.001), whereas the microsphere material (polystyrene or collagen) had no significant effect (Figure 7B).

### 3.8. Multilineage Differentiation Assay

Based on lineage-specific differentiations, hASCs that were cultured on polystyrene and collagen microspheres displayed higher levels of adipogenic and osteogenic differentiation compared to hASCs cultured in polystyrene dishes (controls) using intracellular lipid content (adipogenic) and calcium deposition (osteogenic) assessments (Figure 8). However, while hASCs cultured on collagen microspheres did display higher levels of chondrogenic differentiation compared to controls, those cultured on polystyrene microspheres showed similar chondrogenic differentiation compared to controls using an assessment of cartilage–micromass diameters.

## 4. Discussions

Through novel advances in cell biology, adult stem cells (including hASCs) have been shown to be truly heterogeneous, with subpopulations of cells that may offer potentially greater translational benefits. SSEA-3(+) Muse cells, a representative type of highly functional mesenchymal stem cell, are known to show superior multipotency compared to regular stem cells [32]. However, although routine two-dimensional cultures allow hASCs to expand easily, these conditions adversely affect the proliferation of cells with higher multipotent properties, reducing the usual proportion of SSEA-3(+) hASCs to only approximately 1% [19]. Growing evidence suggests that mechanical and environmental stress greatly influences proliferation, self-renewal, differentiation, and the multipotency of stem cells [33,34]. Among possible sources of stress, gravity may historically have been the most underappreciated microenvironment stress, because of its constant influence on cultured cells.

Numerous studies have reported the impact of microgravity conditions on stem-cell proliferation [35] and differentiation [36,37]. Notably, human mesenchymal stem cells cultured in a microgravity environment usually demonstrate suppressed differentiation [38]. Therefore, some highly functional stem cells may be able to retain their undifferentiated, unaltered phenotypes if they can avoid the force of gravity during culture. According to previous studies, controlled rotation around two axes can minimize the cumulative gravity vector to an average of 10^−3^× *g* over time [24]. In our experimental system, although cells were attached to microsphere surfaces, the culture medium containing the microspheres was stirred by controlled rotation around two axes. As a result, the cells’ gravity vectors were randomly and continuously changing, with their final cumulative gravity vectors being minimized toward zero. Some microgravity-simulating devices have been developed [39,40], but they involve significant costs and are very difficult to standardize. Moreover, even such devices created large cell clusters which represent the limit of cell-proliferative capacity [25] and reinforce the advantage of the present adhesive conditions for the large-scale expansion of hASCs [41].

In previous studies, micro-scale carriers fabricated from a variety of natural and synthetic polymers have shown significant potential for cell growth, specific differentiation, and the delivery of hASCs [42,43]. However, to the best of our knowledge, no investigation has been carried out on the feasibility of culturing cells using stirred microspheres for the stated purpose of mimicking a microgravity environment. The outcomes of this study exceeded our expectations. Seven days of polystyrene and collagen microsphere cultures under static conditions produced an increased number of SSEA-3(+) hASCs compared to polystyrene dish cultures, and those under dynamic conditions exhibited a much higher number of SSEA-3(+) hASCs. These results clearly revealed a positive influence of microspheres on the selective proliferation of SSEA-3(+) cells, especially in those experiments combined with a microgravity environment.

By comparison, our stress-exposure treatments were not very suitable for the expansion of SSEA-3(+) hASCs. Heneidi et al. reported that Muse cells could be isolated from lipoaspirates using severe cellular stress, such as long-term exposure to collagenase [44]. We also confirmed that exposure to proteolysis produced the best results among our stress conditions, increasing the proportion of SSEA-3(+) hASCs by 3.3-fold (4.2%) compared to monolayer cultures (1.3%). During preliminary studies (unpublished data) to enhance the ratio of SSEA-3(+) hASCs (known to be stress-tolerant [19,32]), stress magnitude was carefully optimized in each experiment. However, because 37.6% of viable cells were lost, the final number of proteolytic solution-treated SSEA-3(+) hASCs was only 1.3-fold more (1.7 × 10^3^) compared to the controls (1.3 × 10^3^), with no statistically significant difference. These results suggest that exposure to severe stress can concentrate SSEA-3(+) hASCs through the selective depletion of SSEA-3(−) cells, but such stress can also inhibit cell proliferation. Instead of concentration, microsphere cultures under microgravity conditions increased SSEA-3(+) hASCs by proliferation (Figure 9). In other words, the present microgravity culture conditions increased not only the proportion, but also the number, of SSEA-3(+) cells to actualize regenerative therapies requiring these highly therapeutic cells. However, a limitation of this study is that we only assessed SSEA-3 as a pluripotent marker in stress-exposure experiments. Additional research including analyses of the other pluripotent markers (e.g., Oct4 and Sox2) may further emphasize the differences between microgravity cultures and stress-exposure cultures, or reveal new characteristics of the two cultures.

Furthermore, hASCs harvested from microgravity–microsphere cultures had higher expression levels of the pluripotent markers *OCT4*, *SOX2*, *NANOG*, *MYC*, *KLF*, and *CD34* compared to hASCs cultured in monolayers. *OCT4* and *NANOG* are known to be involved in the self-renewal of human embryonic stem cells, while *SOX2*, *MYC*, and *KLF* expression act as transcription factors to control genes involved during embryonic development [19]. CD34 has been reported to contribute to the proliferative ability of hASCs both by us [23,45] and by others [46,47]. However, the upregulation of nuclear stem-cell markers was hardly detected in hASCs from static microsphere cultures, suggesting that the microgravity environment may play a key role in preserving hASC functionality. In a previous report, spaceflight experiments demonstrated that altered gene expression was associated with epigenetic changes, such as chromatin re-modeling and DNA methylation [48]. The present results extend these observations by demonstrating improved proliferation, angiogenesis, and differentiation into mesenchymal lineages (adipocytes, osteocytes, and chondrocytes) in hASCs from microgravity–microsphere cultures compared to hASCs from cultures under normal conditions. However, the fold change of the genes analyzed by RT-PCR was not so large in this study (less than 1.5), and future studies are required for further qualification of these genes through optimization of the microgravity culture protocols.

Interestingly, the efficacies for collagen and polystyrene microspheres to select for proliferation in highly functional hASCs were roughly equivalent in the present study. Polystyrene microbeads require a plasma coating prior to use to enable good cell attachment; therefore, collagen microbeads may be the superior practical choice for regenerative therapy applications. In addition, as a biodegradable biomaterial, collagen has the advantage that such beads may be clinically administered into a target organ together with hASCs; it may even augment hASC survival and functionality after transplantation by acting as a biological cell scaffold [1,42].

## 5. Conclusions

We cultured hASCs with microspheres using three-dimensional movement to create microgravity conditions for the preservation of stemness, while at the same time preserving cell–microsphere adhesion to maximize cell quantities. This new cell-culturing technique selectively enhanced the proliferation of highly functional stem cells, demonstrating superior stemness, proliferation, differentiation, and angiogenic abilities, which may provide for enhanced treatments for intractable human diseases compared to current hASC treatments. However, further studies are required to optimize a culture protocol to maximize the selective expansion efficiency and to investigate the therapeutic value of these cells using in vivo experiments before clinical trials can proceed.

## Figures and Tables

**Figure 1 cells-10-00560-f001:**
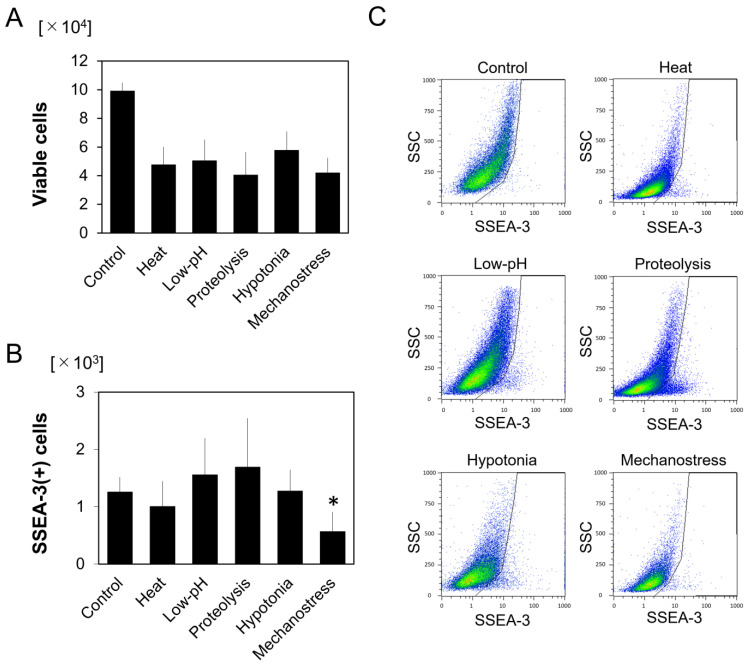
Characterization of SSEA-3(+) human adipose-derived stem cells (hASCs) after exposure to various stress conditions (*n* = 12). (**A**) Numbers of viable hASCs one day after stress exposure. (**B**) Number of SSEA-3(+) hASCs one day after stress exposure. There were no statistically significant differences between stressed cells and normal cells, with the exception of the mechanostress group which showed a significantly lower number of SSEA-3(+) cells (* *p* < 0.01). (**C**) Representative plots of flow cytometry data.

**Figure 2 cells-10-00560-f002:**
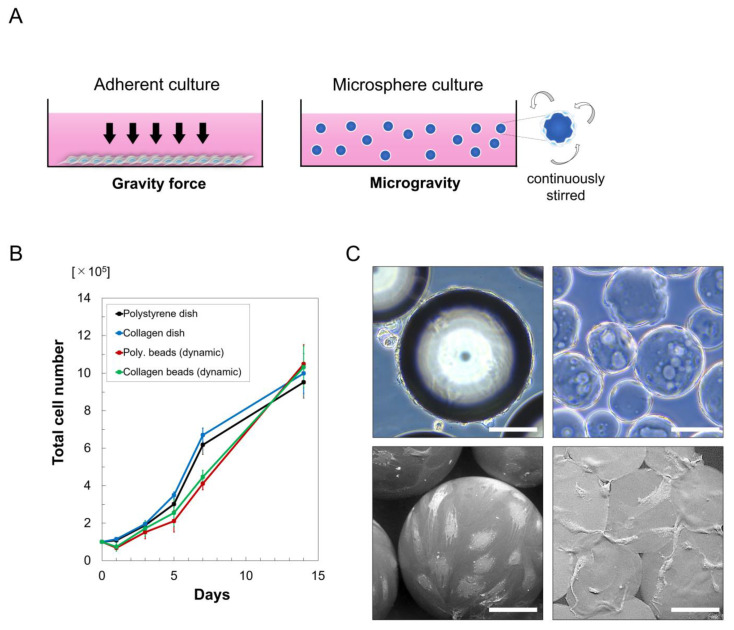
Cell growth in cultures with microgravity and microspheres. (**A**) Schematic representation of the comparison between an adherent culture and a microgravity/microsphere culture. (**B**) Growth curves of human adipose-derived stem cells (hASCs) cultured on polystyrene and collagen microspheres, or in polystyrene and collagen dishes (*n* = 6). Poly, polystyrene. (**C**) Microscopic views of culture-expanded hASCs on polystyrene (upper left) and collagen (upper right) microspheres and scanning electron microscope images of hASCs on polystyrene (lower left) and collagen (lower right) microspheres on day 7. Scale bars = 100 μm.

**Figure 3 cells-10-00560-f003:**
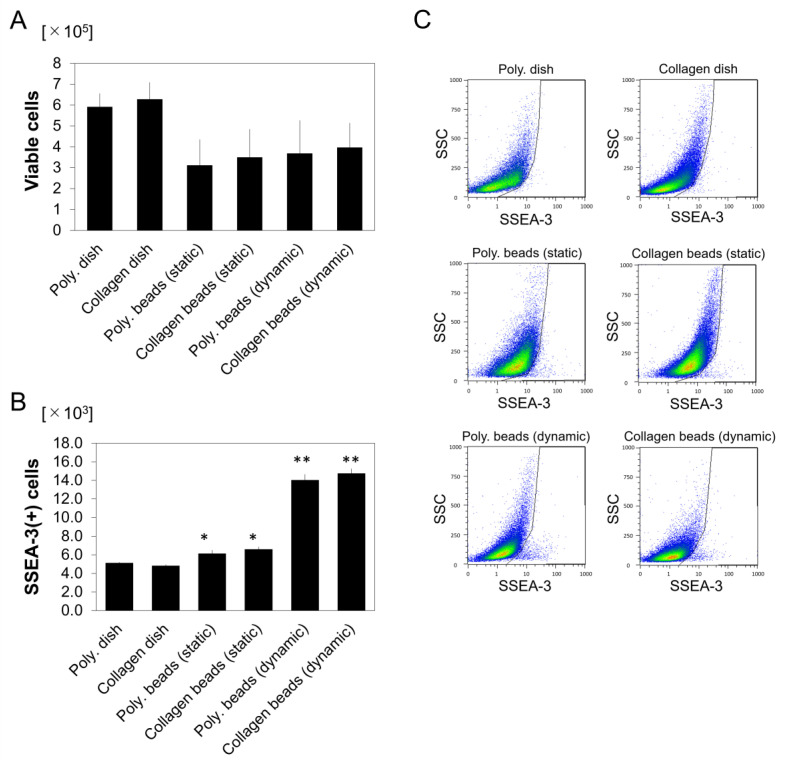
Cellular analyses of human adipose-derived cells after seven days of culture under each of the six conditions (*n* = 6). (**A**) Number of viable cells. Poly, polystyrene. (**B**) Number of viable SSEA-3(+) cells. * *p* < 0.01, ** *p* < 0.001 vs. polystyrene dish (control). (**C**) Representative plots of flow cytometry data.

**Figure 4 cells-10-00560-f004:**
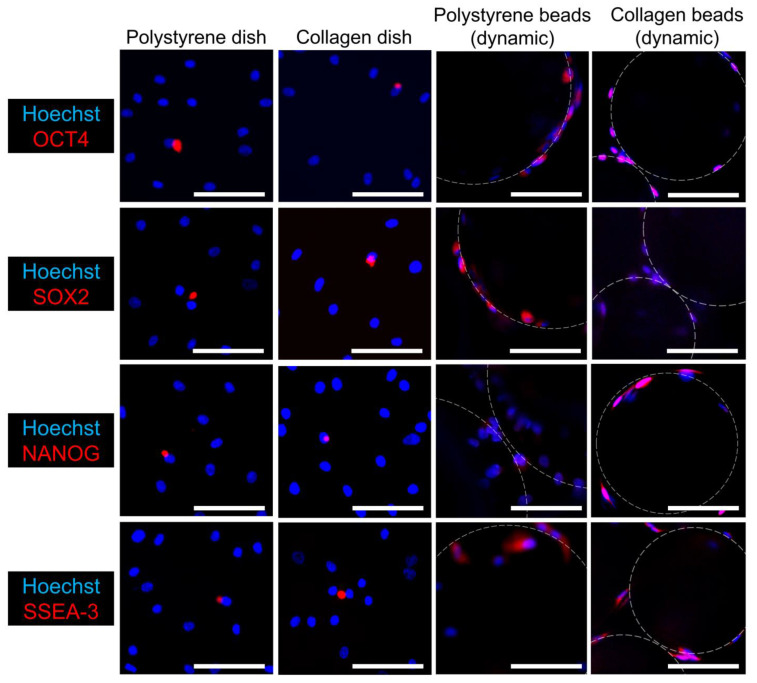
Immunocytochemistry of human adipose-derived cells after seven days of culture. Cells were assessed using immunocytochemistry for the pluripotent markers using 30 μm Z-stack fluorescence images. Broken white lines indicate the outlines of microbeads. Scale bars = 100 μm.

**Figure 5 cells-10-00560-f005:**
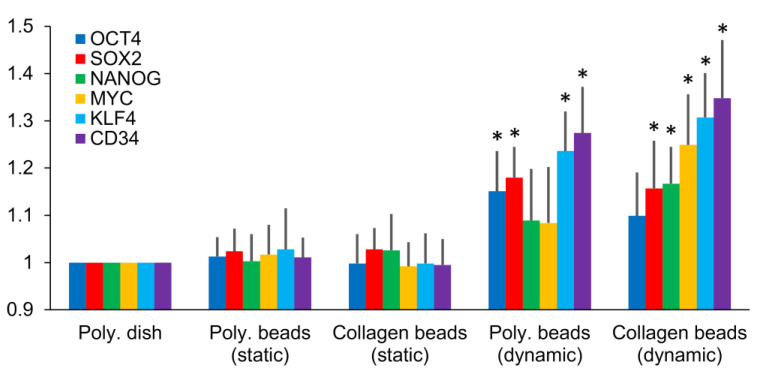
RT-PCR analysis of stem-cell marker expressions in human adipose-derived stem cells on day 7 of various bead-culture conditions compared to marker expressions in cells cultured in normal polystyrene dishes. Poly, polystyrene. * *p* < 0.05 compared to polystyrene dishes.

**Figure 6 cells-10-00560-f006:**
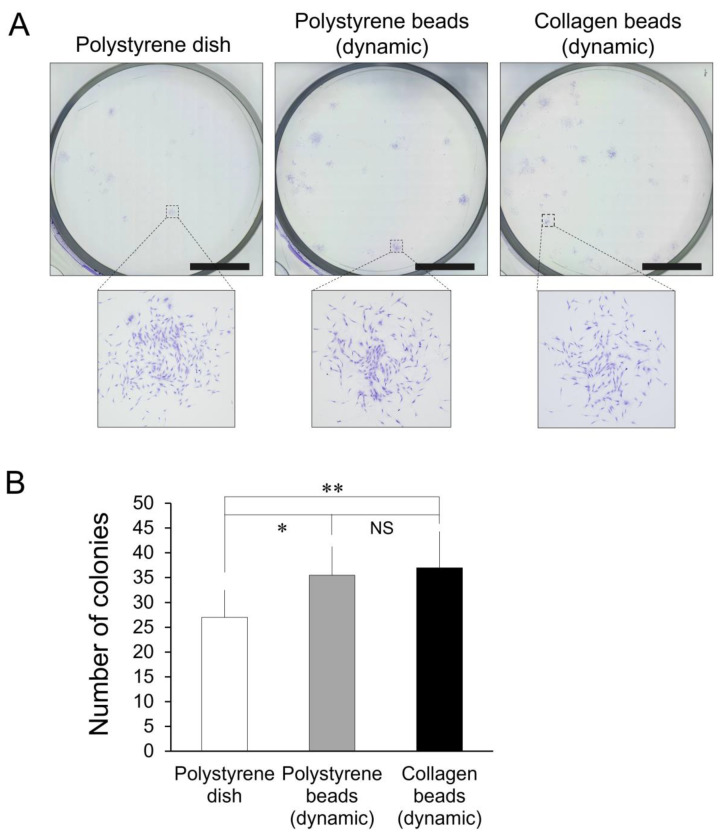
Colony-forming assay results from human adipose-derived stem cells seeded initially 1 × 10^2^ per well. (**A**) Microscopic colonies were stained with cresyl violet after ten days in culture. Scale bars = 1 cm. (**B**) Colony numbers for bead conditions and a non-bead condition (*n* = 6). * *p* = 0.027, ** *p =* 0.025 vs. the polystyrene-dish group.

**Figure 7 cells-10-00560-f007:**
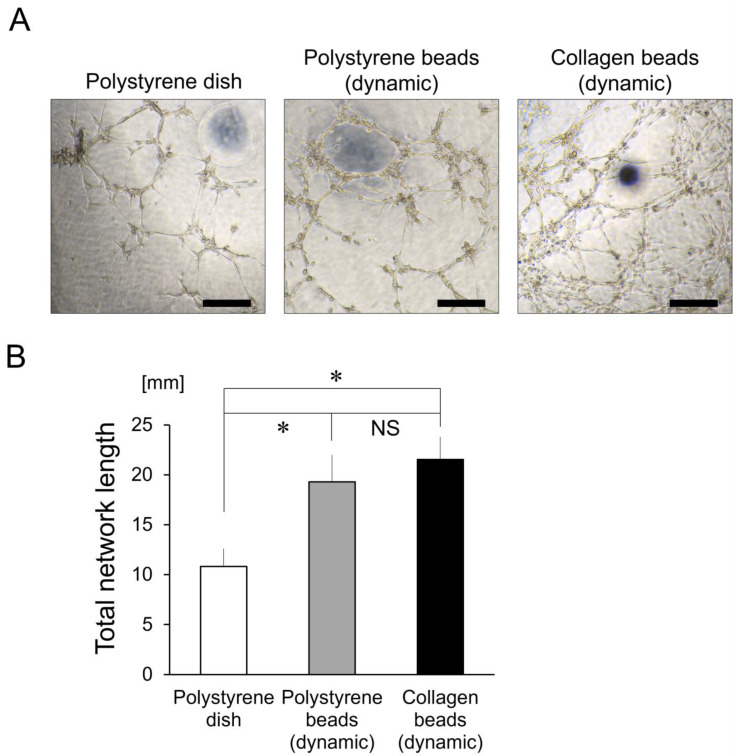
Network-forming assays to test for vasculogenic ability. (**A**) Microscope images from in vitro angiogenesis assays. Scale bars = 300 μm. (**B**) Total network lengths of cytoplasmic extensions were measured (*n* = 6). * *p <* 0.001 vs. the polystyrene-dish group.

**Figure 8 cells-10-00560-f008:**
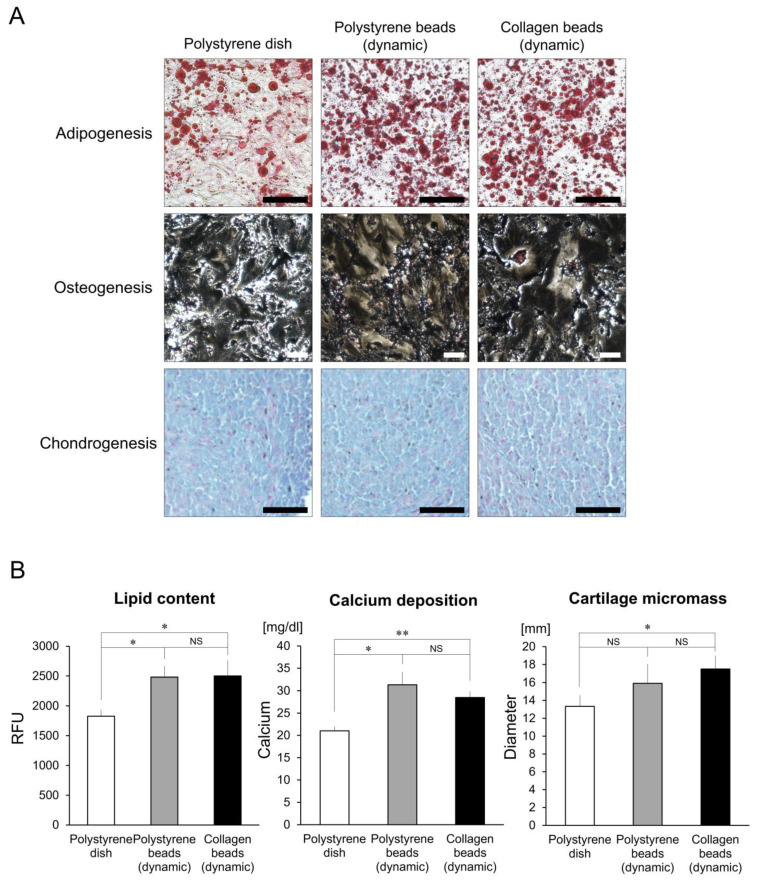
Human adipose-derived stem cell capacity for multilineage differentiation under different culture conditions. (**A**) Microscope images of differential inductions toward adipocytes (oil red O stain), osteocytes (von Kossa stain) and chondrocytes (Alcian blue stain). Scale bars = 100 μm. (**B**) Quantitative analysis of cellular differentiations. The capacities for multilineage differentiations are indicated by measurements of accumulated lipid (adipogenesis); calcium deposition (osteogenesis); and micromass diameters (chondrogenesis). * *p <* 0.05 vs. polystyrene dishes, ** *p <* 0.005 vs. polystyrene dishes.

**Figure 9 cells-10-00560-f009:**
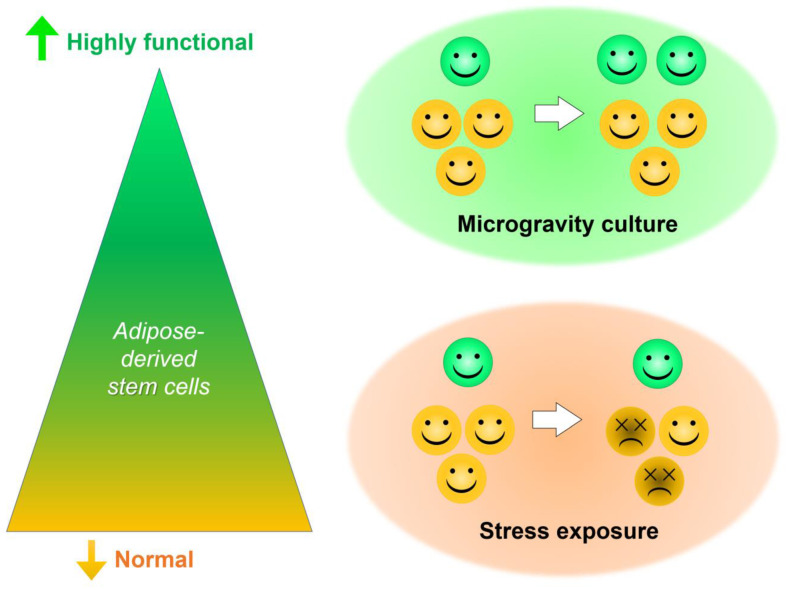
Schematic representation of the study. Human adipose-derived stem cells are normally capable of heterogeneous potency. Exposure to stress can selectively deplete regular stem cells, but does not increase the number of highly functional stem cells. In contrast, microgravity/microsphere cultures can selectively cause the proliferation of highly functional cell subpopulations to reach the numbers required for regenerative therapies.

**Table 1 cells-10-00560-t001:** Primer sequences used for real-time PCR.

Gene	Primer Sequence (5′–3′)	
*ACTB*	Forward:	TGAAGTGTGACGTGGACATC
	Reverse:	GGAGGAGCAATGATCTTGAT
*OCT4*	Forward:	AGCGAACCAGTATCGAGAAC
	Reverse:	TTACAGAACCACACTCGGAC
*SOX2*	Forward:	AGCTACAGCATGATGCAGGA
	Reverse:	GGTCATGGAGTTGTACTGCA
*NANOG*	Forward:	TGAACCTCAGCTACAAACAG
	Reverse:	TGGTGGTAGGAAGAGTAAAG
*MYC*	Forward:	ACTCTGAGGAGGAACAAGAA
	Reverse:	TGGAGACGTGGCACCTCTT
*KLF4*	Forward:	TCTCAAGGCACACCTGCGAA
	Reverse:	TAGTGCCTGGTCAGTTCATC
*CD34*	Forward:	CCTCAGTGTCTACTGCTGGTCT
	Reverse:	GGAATAGCTCTGGTGGCTTGCA

*ACTB*, β-actin; *OCT4*, octamer-binding transcription factor 4; *SOX2*, (sex determining region Y)-box 2; *KLF-4*, Kruppel-like factor 4.

**Table 2 cells-10-00560-t002:** The six culture conditions for human adipose derived stem cells. The data represent cells after culture for seven days (*n* = 6).

	Cell Number	Cell Viability	SSEA-3 Positivity	SSEA-3(+) Cell Growth Rate
Polystyrene dish	(5.9 ± 0.7) × 10^5^	99.8 ± 0.2%	1.1 ± 0.4%	5.1
Collagen dish	(6.3 ± 0.8) × 10^5^	99.9 ± 0.1%	1.0 ± 0.5%	4.8
Polystyrene beads (static)	(3.8 ± 1.2) × 10^5^	81.3 ± 7.7%	2.5 ± 0.9%	6.1
Collagen beads (static)	(3.9 ± 1.4) × 10^5^	89.0 ± 9.2%	2.4 ± 1.1%	6.6
Polystyrene beads (dynamic)	(3.8 ± 1.6) × 10^5^	95.8 ± 3.6%	4.8 ± 1.5%	14.0
Collagen beads (dynamic)	(4.0 ± 1.1) × 10^5^	98.9 ± 1.0%	4.7 ± 1.4%	14.7

SSEA-3(+) cell growth rate = SSEA-3(+) cell number on day 7/day 0.

## Data Availability

All relevant data are within the manuscript.

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
