# Peer review of "Selective Proliferation of Highly Functional Adipose-Derived Stem Cells in Microgravity Culture with Stirred Microspheres"

_cells, 2021, doi:10.3390/cells10030560_

Round 1

Reviewer 1 Report

Dear Authors,

The article " Selective proliferation of highly functional adipose-derived stem cells in microgravity culture with stirred microspheres" is nicely executed article. After minor language improvement, the manuscript can be accepted for publication.

Regards,

Author Response

We thank the reviewer’s comment. This manuscript has received further professional linguistic review by a native English speaker experienced in editing scientific papers.

Reviewer 2 Report

Authors claimed that they invented new cell-culturing technique selectively enhanced the proliferation of highly functional stem cells. To this end, they used microbeads supporting 3D movement to create microgravity conditions for the cells. I totally agree that culture technique must be developed for the stem cell therapy to enhance its therapeutic ability. However, experimental design should be improved to support their conclusions.

1. I wonder that it is really microgravity condition using 3D beads. Infect, cells attached to surface of beads is not exposured to microgravity which cells suspended in media without any attachment.

2. I cannot understand that SSEA positive cells did not show that changes of viability under various stress and if it is true, It seems that the development technique to enhance cell viability is not necessary. (Fig. 1)

3. The characterization of firstly isolated ADSC should be provided addition to Fig. 1.

4. Fig. 4. It is need to show high quality image picture to support authors claim. It looks like that dramatically differences between dishes and bead unlike the result of qRT-PCR (Fig.5) which not more than 2 fold differences.

Reviewer 3 Report

Dear authors

Your paper reports the preservation of stemness of human adipose-derived stem cells (hASCs), using microgravity conditions with microspheres (size of 100−300 μm) in stirred suspension. Indeed, one week of cultures with polystyrene and collagen microspheres increased the proportion of SSEA-3(+) hASCs by 4.4 and 4.3 times (2.7 and 2.9 times their numbers), respectively, compared to normal culture conditions. The cultured hASCs expressed higher levels of pluripotent markers (OCT4, SOX2, NANOG, MYC, and KLF), and improved abilities for proliferation, colony formation, network formation and multiple mesenchymal differentiation. 

I believe that your paper is interesting and can be accepted with minor revision.

minor revision:

  • insert a characterization of hASCs using specific mesenchymal markers like CD73, CD105, CD90, etc. Furthermore, it would be interesting to test the typical mesenchymal stemness markers after cultivation with polystyrene and collagen microspheres.
  • completely review the bibliography according to the instructions for the authors.                                                                                          Journal Articles: 1. Author 1, A.B.; Author 2, C.D. Title of the article.    Abbreviated Journal Name Year, Volume, page range.

Round 2

Reviewer 2 Report

The authors should at least place more data for Figure 1 and Figure 4 to clearly support their conclusions. But these data still have not improved.

In addition, I disagree with the explanation of Figure 5. The authors concluded that the system significantly increased stamen-related genes, but the fold change of the genes appeared to be small, such as less than 1.5 times.

I regret that I cannot be more active in this situation. However, there must be more qualified data to improve this article. 

Round 3

Reviewer 2 Report

None.